ecology

climate change, marine fish communities, assembly processes, functional variance, community-weighted trait variance, biodiversity

**Author for correspondence:**
André Frainer
e-mail: andre.frainer@nina.no

# Increased functional diversity warns of ecological transition in the Arctic

André Frainer[1,2], Raul Primicerio[1], Andrey Dolgov[3,4,5], Maria Fossheim[6], Edda Johannesen[6], Sigrid Lind[7] and Michaela Aschan[1]

[1]Norwegian College of Fishery Science, UiT The Arctic University of Norway, 9037 Tromsø, Norway
[2]Norwegian Institute for Nature Research (NINA), Fram Centre, 9007 Tromsø, Norway
[3]Polar Branch of Russian Federal Research Institute of Fisheries and Oceanography, 183038 Murmansk, Russia
[4]Murmansk State Technical University, 183010 Murmansk, Russia
[5]Tomsk State University, 634050 Tomsk, Russia
[6]Institute of Marine Research (IMR), Fram Centre, 9007 Tromsø, Norway
[7]Norwegian Polar Institute (NPI), Fram Centre, 9007 Tromsø, Norway

AF, 0000-0002-3703-7152; AD, 0000-0002-4806-3284

As temperatures rise, motile species start to redistribute to more suitable areas, potentially affecting the persistence of several resident species and altering biodiversity and ecosystem functions. In the Barents Sea, a hotspot for global warming, marine fish from boreal regions have been increasingly found in the more exclusive Arctic region. Here, we show that this shift in species distribution is increasing species richness and evenness, and even more so, the functional diversity of the Arctic. Higher diversity is often interpreted as being positive for ecosystem health and is a target for conservation. However, the increasing trend observed here may be transitory as the traits involved threaten Arctic species via predation and competition. If the pressure from global warming continues to rise, the ensuing loss of Arctic species will result in a reduction in functional diversity.

## 1. Introduction

In the Arctic, shifts in species distribution driven by climate warming are rapidly changing the composition of marine fish communities [1–4]. Whereas marine species are lost at higher rates in the tropics [5–7], they are gained in the Arctic due to southerly species range expansion [2,3,8]. With Arctic warming, an increase in water temperature and sea-ice loss [9] induce changes in habitat and resource availability [10], favouring the colonization by species that differ in resource requirement and traits from the endemic species [1,3]. As the Arctic communities have low functional diversity, incoming species are expected to broaden the distribution of functional traits, increasing functional diversity [11] and influencing adaptive capacity and ecosystem functioning. However, shifts in species range and changes in habitat conditions may trigger local extinctions of Arctic species due to ecological interactions and habitat loss, with subsequent reduction in functional diversity. The net effect of warming on Arctic functional diversity is, therefore, unknown.

The northern Barents Sea is one of the most rapidly warming places on Earth [12], with the strongest temperature increases of the lower atmosphere, the greatest winter sea-ice loss and a rapidly warming ocean [9], hence it is a sentinel region for climate change. Like other Arctic marine ecosystems, it is cold, stratified and ice covered. It has a colder, fresher Arctic water layer that protects the sea-ice cover from a deeper layer of warm and saline Atlantic water. But, after approximately 2005, the stratification has weakened, resulting in a dramatic warming of the Arctic layer and rapidly diminishing sea-ice cover [9], modifying the habitat characteristics relevant for the demersal fish community and leading to its reorganization.

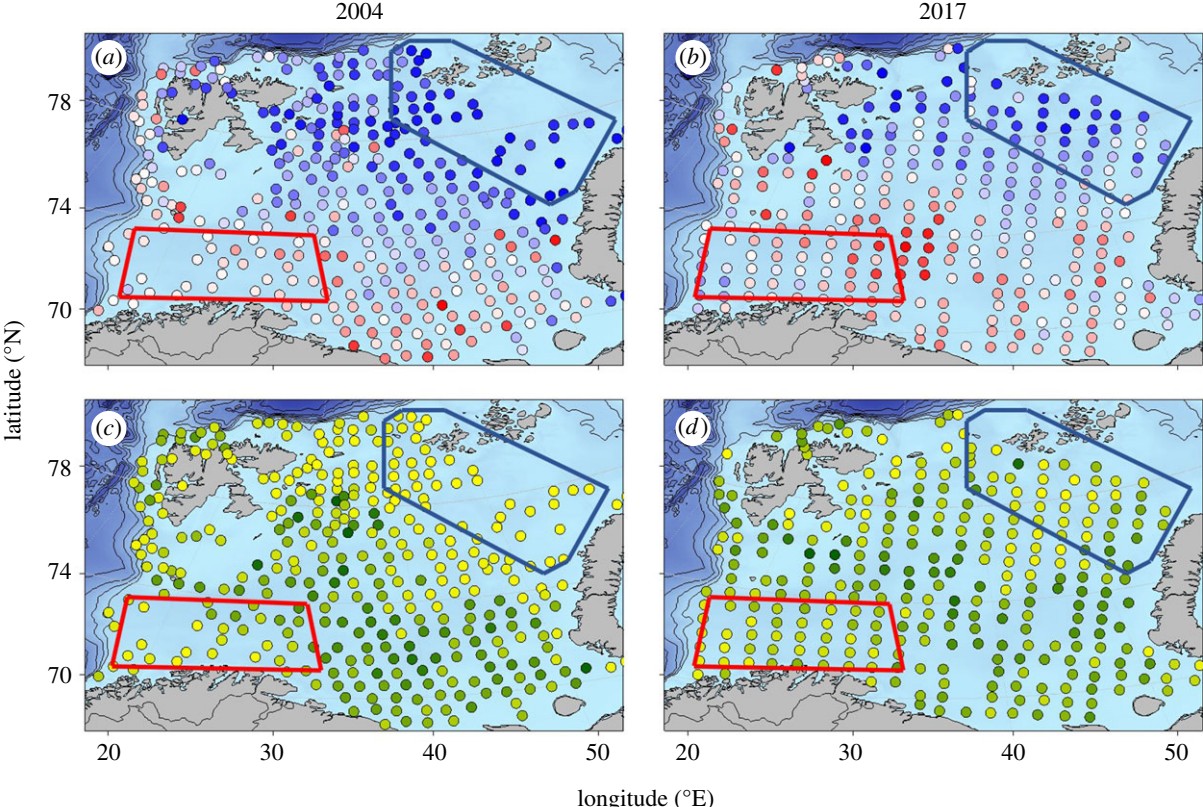

**Figure 1.** Change in ecosystem functional characterization in the Barents Sea. Community-weighted mean trait values (CWM) (*a,b*) and community-weighted trait variance (CWV) (*c,d*) estimated from observations of demersal fish during ecosystem surveys in two subregions of the Barents Sea (blue, Arctic; red, boreal), in the first and last years of the ecosystem survey data, 2004 (*a,c*) and 2017 (*b,d*). Circles refer to sampled sites and are coloured in two colour gradients. For CWM, colours range from blue, indicating dominance of Arctic-like traits, to red, indicating dominance of boreal-like traits. For CWV, colours range from yellow, indicating low trait variance (low heterogeneity), to green, indicating high trait variance (high heterogeneity). (Online version in colour.)

Here, we use an extensive dataset of over 4000 stations sampled during 2004–2017 [13] to investigate changes in fish biodiversity and apply a community-wide multi-trait analysis that allows the assessment of the magnitude and character of functional diversity. The high rates of environmental and ecological change experienced in this Arctic warming hotspot—one of the best-observed marine ecosystems in the world—provide a unique opportunity to investigate the influence of climate change on assembly processes and their ecosystem implications.

## 2. Material and methods

The taxonomic and functional properties of fish communities were assessed based on 14 years of detailed information on the abundance of 49 demersal fish species found in the Barents Sea. Sampling was done using standardized shrimp bottom trawls (Campelen 1800) at every 60 km (35 nm, towed at approx. 3 km for 15 min) and covered the entire shelf sea (surface area approx. 1 400 000 km$^2$) from the boreal region north of Norway and the Kola peninsula in Russia, to the Arctic archipelagos of Svalbard, Franz Josef and Novaya Zemlya, totalizing 4223 sampled fish communities [13].

Comprehensive information of 15 functional traits, including habitat affinity, life history, body size, feeding ecology and food-web characteristics [1] were used to estimate functional characterization and complementary measures of functional diversity—functional richness, functional dispersion and functional variance (electronic supplementary material, table S1). The chosen traits provide information on species characteristics that are involved in assembly processes (life history, habitat

affinity and feeding ecology), and ecosystem functions (feeding ecology and foodweb characteristics) [14]. Together with taxonomic diversity, the above-mentioned functional diversity indices provide an assessment of magnitude and character of changes in biodiversity. Species richness is the count of unique species found in each sampling location (individual hauls) and species evenness was measured following Pielou's index, as: Shannon index/(log(richness) + 1). Functional richness, calculated with the R software [15] package *FD* [16], was measured on a dendrogram of fish functional traits as the sum of branch length connecting all species in a community. This measure yields the functional distance between species in a community, and increases as species with differing trait values are added [17]. Functional dispersion, measured as Rao's Q [18], was calculated using R package *FD* [16], and indicates the spread of species in trait space and is affected by both the trait distance and abundance distribution of the species in a community [19,20]. Changes in functional dispersion may be due to the gain or loss of traits or stem from changes in species evenness and richness [20]. Disentangling the relative contributions of functional richness, species evenness and species richness to functional dispersion helps to infer which are the candidate ecological explanations for the observed trends in functional diversity. For this purpose, we used a general mixed effect linear model applied to all sampling locations across the entire Barents Sea, with years as random effect, and spatial-autocorrelation structure nested by region. We further analysed the contribution of the three indices to the trend in functional dispersion within each region using region-specific mixed effect models. Mixed effect models were run with the R package *nlme* [21].

To identify the traits responsible for changes in functional diversity, we computed the community-weighted variance of

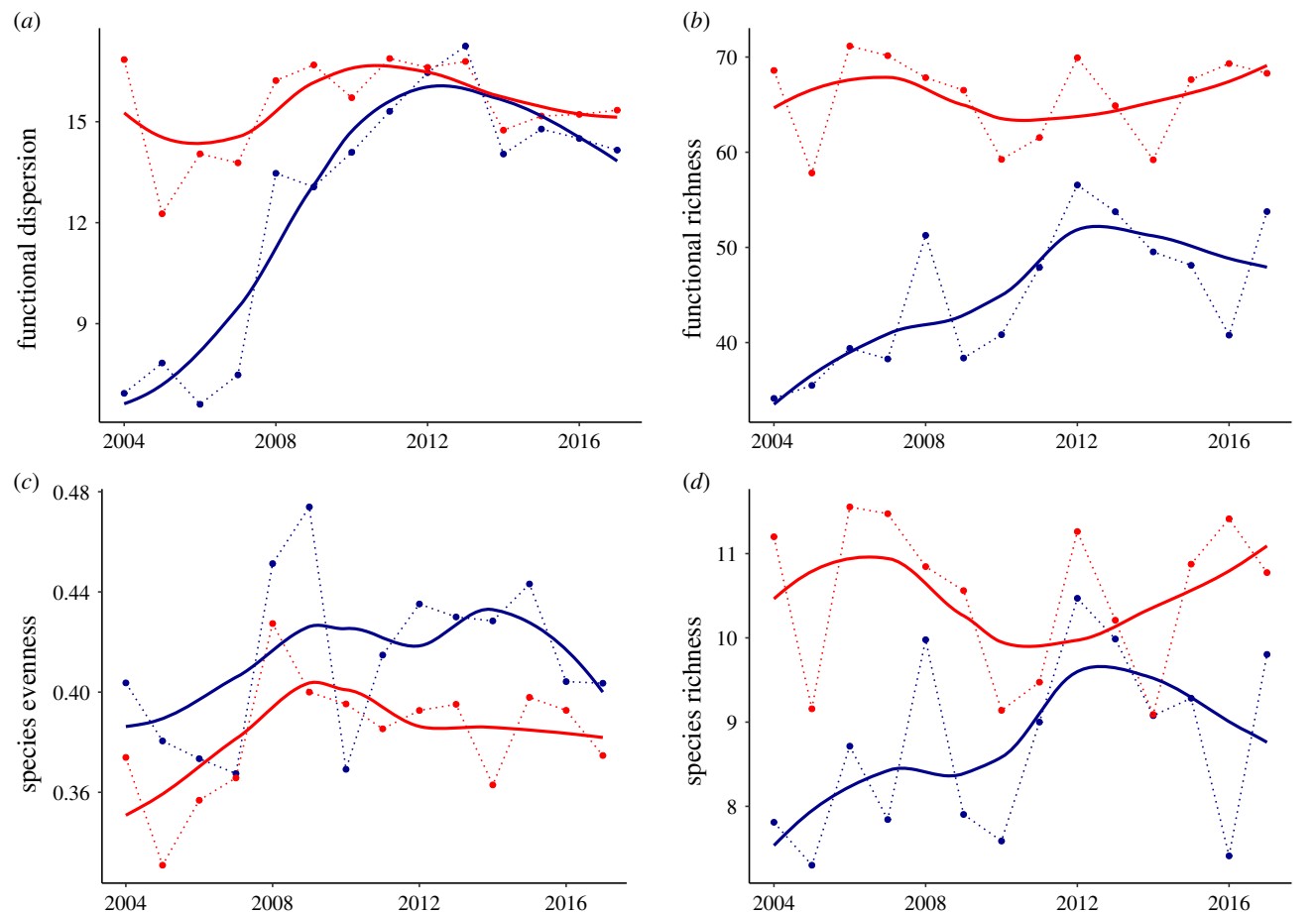

**Figure 2.** Observed changes in fish functional diversity estimated from common diversity measures. Common functional diversity metrics averaged over the Arctic (in blue) and boreal (in red) regions of the Barents Sea (polygons shown in figure 1), showing (*a*) functional dispersion, (*b*) functional richness, (*c*) species evenness and (*d*) species richness. Solid lines are smoothed averages and dotted lines connected the mean values taken after pooling all stations found in each region. (Online version in colour.)

trait values (CWV), using multiple traits simultaneously, thus extending previous single-trait approaches [22]. For this, traits were scaled and fuzzy coded and the multivariate analysis was done with the R package *vegan* [23]. The variance of each individual trait within sampling units was weighted by the relative abundance of all species characterized by that trait. We then investigated the CWV of the 15 traits simultaneously by computing a principal component analysis (PCA). The principal components of the CWV PCA provide the basis for metrics of *functional variance*, allowing the simultaneous assessment of the magnitude (by means of the CWV scores on the principal components) and character (by inspecting the CWV factor loadings on the principal components) of functional variance. Thus, the functional variance approach proposed here provides a good candidate metric and can replace traditional indices of functional diversity (e.g. functional dispersion, functional richness or functional evenness) that lack information on which functional traits are responsible for the functional diversity. Hence, we name the values obtained from the first principal component as *functional variance PC1*, to distinguish them from the other components. We also extended previous analyses of community-weighted mean trait values (CWM) [1] to include data from 2014 to 2017. Together, the main components of the CWV and CWM PCA provide a summary of functional traits distributions. The R script for calculating CWV for multiple traits simultaneously is provided in the electronic supplementary material.

We analysed the temporal development of biodiversity and functional traits related to key ecosystem functions (see more explanation in the Results and discussion section) in two distinct

climatic and zoogeographic regions of the Barents Sea, a northeastern Arctic and a southwestern boreal region (figure 1). For these analyses, we use the mean values within each region and year (figures 2 and 3), but we provide a complementary analysis of the temporal development of functional variance using all individual sampling stations in each region, corrected for spatial autocorrelation, in the electronic supplementary material. For taxonomic diversity indices, variation in sampling effort within regions may bias the estimates. The two regions have good sampling coverage during the study period with the exception of the low sampling effort in 2014 in the Arctic region, caused by extensive sea-ice coverage limiting survey sampling. The regions have been analysed in previous publications for species distribution [2], food web metrics [24] and CWM values [1], providing important ecological background information for this study. We also map changes in biodiversity across the entire Barents Sea (electronic supplementary material, figures S1 and S2).

## 3. Results and discussion

We found that functional dispersion in the Arctic region was low in 2004, but increased rapidly to levels comparable with the boreal region by 2012, maintained thereafter until at least 2017 (figure 2*a*). This major increase in functional dispersion in the Arctic and its levelling with the boreal region is unprecedented and predates by several years the expected convergence in CWM of Arctic and boreal regions based on

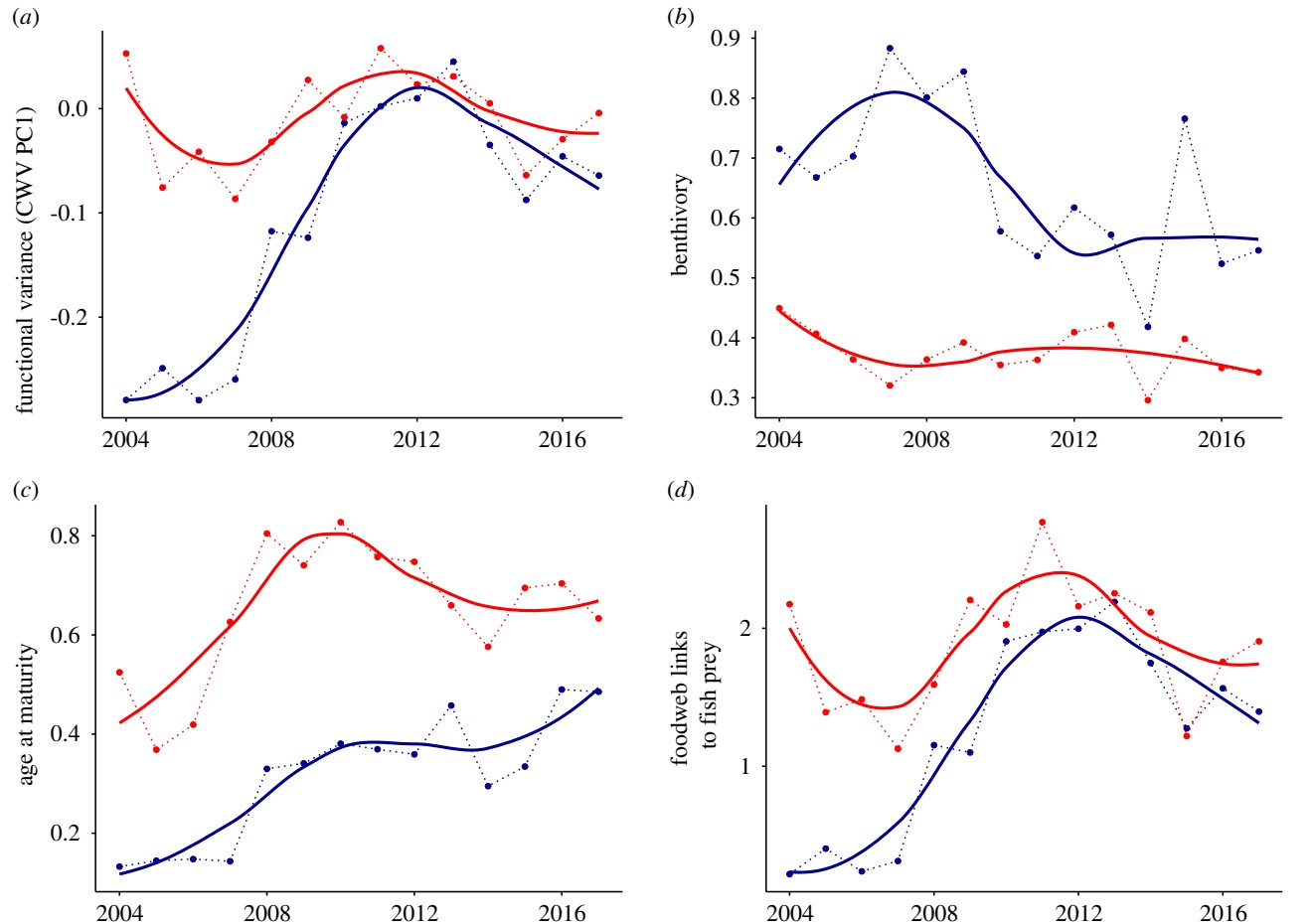

**Figure 3.** Observed changes in the variance of fish functional traits. Mean CWV values pooled in the Arctic (in blue) and boreal (in red) regions of the Barents Sea (polygons shown in figure 1). (*a*) *Functional variance*, measured as the first principal component of the community-weighted trait variance analysis; (*b–d*) the variance of individual traits in each region: community-weighted variance of (*b*) benthivory, (feeding ecology) (*c*) age at maturity (life history) and (*d*) foodweb links to fish prey (foodweb-related characteristics). Solid lines are smoothed averages and dotted lines connected the mean values taken after pooling all stations found in each region. CWV of all functional traits is shown in electronic supplementary material, figure S3. (Online version in colour.)

recent estimates [1]. The increase in functional dispersion may be due to the addition of novel boreal traits or stem from changes in species evenness in the Arctic region. Together, species richness (slope: $-0.29 \pm 0.01$, $p < 0.001$), species evenness (slope: $7.02 \pm 0.1$, $p < 0.001$) and functional richness (slope: $0.06 \pm 0.001$, $p < 0.001$) accounted for 58% of the spatio-temporal variation in functional dispersion across the Barents Sea (electronic supplementary material, figure S3). Region-specific analyses indicate that the three variables (species richness, species evenness and functional richness) explain the variance in functional dispersion better in the boreal than in the Arctic region ($r^2 = 0.78$ and 0.50, respectively). Thus, although functional richness, species richness and evenness increased faster in the Arctic than in the boreal region, their rate of increase does not explain the higher rate of change in functional dispersion observed in the Arctic (figure 2a–d).

Our approach using CWV on multiple traits can help explain the rapid increase in functional dispersion seen in the Arctic. The first principal component (PCI) of the multiple-trait CWV—our *functional variance PC1*—accounted for 56% of variation in the data (electronic supplementary material, figure S4). The functional variance PC1 was a good indicator of functional dispersion, being highly correlated with that metric ($r^2 = 0.83$). The functional traits associated with the functional variance PC1 were foraging-

related traits, specifically traits that describe the number of trophic links to fish prey, and to bird and mammal predators (electronic supplementary material, figure S5). Functional traits that describe somatic growth capacity and temperature affinity were also positively associated with functional variance PC1. The variance of functional traits associated with PC1 was lower in the Arctic than in the boreal region in 2004, but increased rapidly in the Arctic region, converging to boreal levels by 2012 (figure 3). The PC1 of the Arctic CWM also increased rapidly, but did not reach boreal values due to the persistence of Arctic species in the region (figure 4). Our CWV analysis of multiple traits provides an alternative to measures of functional dispersion with the added benefit of characterizing the traits involved in changes in functional diversity. Such characterization is crucial to infer assembly processes and understand community responses to environmental change.

The strong increase in functional variance of the Arctic Barents Sea region is driven by the incoming boreal species, whose traits differ from those of Arctic species that still persist in the region (electronic supplementary material, figure S6). These incoming boreal species enter the Arctic region at different times, and the increased importance of boreal traits observed in the Arctic region is due to colonization by several boreal species throughout the study period (electronic supplementary material, figures S5–S9). Our

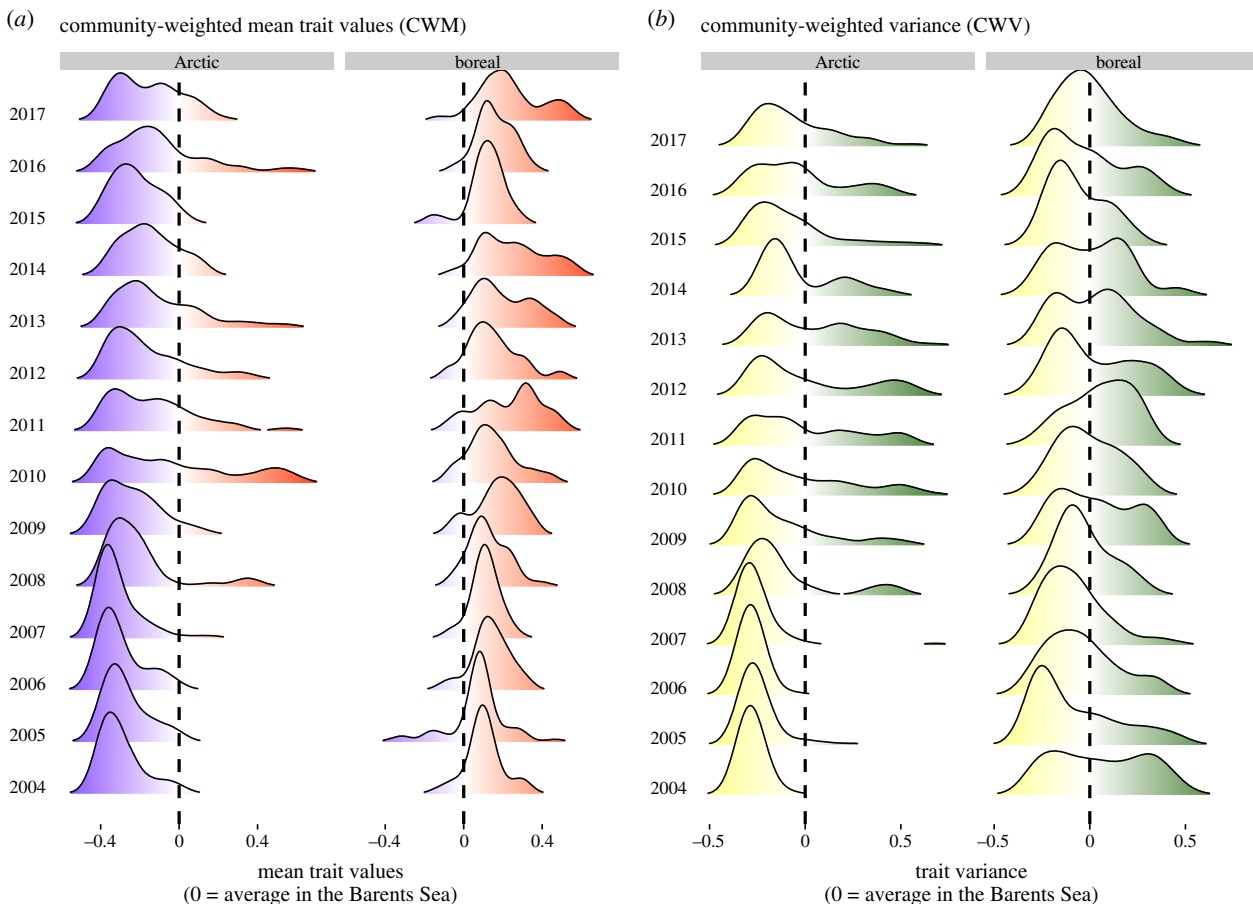

**Figure 4.** Temporal development in the distribution of (*a*) community-weighted mean trait values and (*b*) community-weighted variance in the Arctic and boreal regions of the Barents Sea (polygons shown in figure 1). (Online version in colour.)

findings are consistent with expectations from the community assembly process, which predicts that, in order to establish in the Arctic, boreal species must cope with the environmental conditions and ecological interactions with resident Arctic species [25]. The increasing water temperatures in the northern Barents Sea facilitate the establishment of boreal species, such as cod, haddock and redfish, which are better suited for warmer waters than the cold-adapted Arctic species. Further, the boreal species entering the Arctic differ from the resident species in terms of resource use, by either feeding on different prey or on the Arctic fish species themselves, thereby being able to cope with competition. The fish species typically found in the Arctic are generally smaller and more specialized on benthic invertebrate prey than the incoming boreal species [1,11]. The latter can also feed on pelagic prey, including fish (e.g. polar cod), and are thereby advantaged by the climate-driven increase in pelagic productivity due to poleward expansion of Atlantic waters [26] and reduced Arctic ocean stratification [27]. The primary role of foraging traits in fuelling the increase in Arctic functional variance strongly suggests the importance of ecological interactions in the assembly process determining which incoming boreal species may successfully establish.

The incoming boreal traits can further change ecosystem functions and food web configurations in the Arctic [28]. The fish species moving rapidly northward in the Barents Sea are mostly motile, large, piscivorous generalists. This contrasts with findings from other North Atlantic seas, where

small pelagic fish are observed to move northwards [29–31]. The increase in functional variance promotes adaptive capacity [32] of Arctic communities, but if the trend in climate-driven habitat modification persists, these communities might lose the benthivore fish component, leading to biodiversity loss and, therefore, reduction in adaptive capacity. A homogenization of fish communities is already observed across many parts of the Barents Sea due to the increase in cod abundance, a motile top predator with potential for wide-spread top-down effects [33,34]. As our results show, the Arctic ecosystem is gaining fish species occupying higher trophic levels than the resident ones, and relying both on pelagic and benthic resources. The increased importance of piscivory will trigger top-down effects, reducing the number of secondary consumers and releasing their prey from predation pressure. Thus, the widened resource usage to include pelagic prey will influence the degree of bentho-pelagic coupling and the importance of pelagic productivity in sustaining the fish community in the Arctic.

Our results on functional variance documenting the magnitude and character of recent changes in functional traits provide a measure of the ongoing ecological transition in Arctic marine ecosystems and are an early warning of impact to come if the pressure from global warming continues to rise. Further warming and reduction in sea-ice coverage will most likely continue to negatively affect the Arctic fish species whose food availability largely depends on the coupling between sea-ice cover and the benthic

production [35], and whose vulnerability to natural enemies will be challenged by new incoming predators.

Data accessibility. All data on fish abundance and composition from the Norwegian surveys are freely available from the IMR official repository: https://doi.org/10.21335/NMDC-1657305299. The data on fish abundance and composition from the Russian surveys belong to the Polar branch of VNIRO and cannot be made available, but can be assessed in the joint IMR-PINRO reports (https://www.hi.no/hi/nettrapporter/imrpinro/2018/imr-pinro-report_2-2018). The R scripts used for the analyses of community-weighted trait variance for multiple species and all data on functional diversity are freely available on GitHub repository: https://github.com/andre-frainer/Functional_Variance.git. The full data table with information on all 15 traits from the 49 fish species analysed in our work is available in electronic supplementary material, table S1.

Authors' contributions. A.F. and R.P. designed the study. A.F. analysed the data and wrote the first draft of the manuscript with special contribution from R.P. All authors contributed to the writing.

Competing interests. We declare we have no competing interests.

Funding. Funding was provided through the European Union Horizon 2020 Project ClimeFish 677039 (to A.F., M.A., R.P. and M.F.).

Acknowledgements. We are grateful for the valuable work of technicians and science staff involved in the several ecosystem survey cruises in both Norway and Russia. Two anonymous reviewers provided valuable comments on the manuscript.

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
