## [Peer Review File · Proceedings of the Royal Society B: Biological Sciences]

Review History

RSPB-2020-1458.R0 (Original submission)

Review form: Reviewer 1

Recommendation

Reject – article is not of sufficient interest (we will consider a transfer to another journal)

Scientific importance: Is the manuscript an original and important contribution to its field?

Marginal

General interest: Is the paper of sufficient general interest?

Excellent

Quality of the paper: Is the overall quality of the paper suitable?

Excellent

Is the length of the paper justified?

Yes

Should the paper be seen by a specialist statistical reviewer?

No

Do you have any concerns about statistical analyses in this paper? If so, please specify them explicitly in your report.

Yes

It is a condition of publication that authors make their supporting data, code and materials available - either as supplementary material or hosted in an external repository. Please rate, if applicable, the supporting data on the following criteria.

Is it accessible?

No

Is it clear?

Yes

Is it adequate?

Yes

Do you have any ethical concerns with this paper?

No

Comments to the Author

I have reviewed the manuscript entitled "Increased functional diversity warns of ecological transition in the Arctic" for Proceedings of the Royal Society B. This study is providing evidence for temporal changes in several taxonomic and functional diversity indices in the Barents Sea using scientific surveys and a collection of functional traits. This manuscript is concise and well-written, and it was a pleasure to read the study. This work seems to be a follow up study to a previous publication from 2017 entitled "Climate-driven changes in functional biogeography of Arctic marine fish communities", published in PNAS, and led by the same first author. Although the authors apply multiple diversity indices (functional dispersion, functional richness, species richness and evenness, community-weighted variance), I don't think the overall conclusions and findings are novel compared to the previous study published in 2017. The results from the study in 2017 are essentially shown in Figure 4a. Using community-weighted variance and functional dispersion seem to show similar results (Figure 3a/4b and Figure 2a, respectively). On the contrary to what the authors state (in the abstract and introduction), I don't find the methods employed to assess functional diversity that novel, as they have been developed more than a decade ago (and even more for the taxonomic indices used) and have been very frequently applied in ecology. However, even if not novel, it can of course provide interesting insights in community changes. I further have some comments on the methodology applied in the manuscript. I think the work done is of good quality, but I think the authors could dig in more in their analyses to better understand taxonomic and functional biodiversity dynamics with more insightful indices than richness and evenness. The ecosystem studied by the authors is very well documented and provides very important insights for management and conservation of species under climate change, so I encourage the authors to improve their analyses to further unravel diversity changes in the Barents Sea. I list below a number of points which I hope will help the authors to improve their study (I have not authored any of the literature sources mentioned below):

- The authors mention multiple times that changes in functional diversity will affect ecosystem functioning, and I think the authors could link changes in trait composition/ diversity to ecosystem functioning, by comparing with spatio-temporal changes in biomass/benthopelagic coupling through time. It has been evidenced that climate change is, through an increase in diversity and changes in species habitat use, increasing ecosystem functioning (for instance, Friedland et al., 2020). It would be interesting to see if similar changes occur in the Barents Sea, and they can be attributed to changes in trait composition and taxonomic diversity.

-Friedland, K.D., Langan, J.A., Large, S.I., Selden, R.L., Link, J.S., Watson, R.A., Collie, J.S., 2020.

Changes in higher trophic level productivity, diversity and niche space in a rapidly warming continental shelf ecosystem. *Science of the Total Environment* 704, 135270.

- The case of the Barents Sea is very interesting and the ecosystem is obviously going through major climate-driven changes. However, the trends in taxonomic and functional diversity found by the authors may hide more complex changes. Species richness is a well-known poor indicator to quantify complex biodiversity changes in communities in time and space. While the polar part of the ecosystem is inhabited by new species previously absent, each sub-system might still go through gains and losses of species. It would be interesting to apply methods providing evidence for gains in species and abundance of boreal species, as well as losses/decrease in benthic resident polar species. The authors particularly state that the increase in functional diversity is not beneficial for the ecosystem, but they do not show how unbeneficial it is in terms of local diversity. This is however well done for trait distributions in Figure 4, but I think the authors could go further in their analyses beyond trait distributions. I understand that the arrival of new species could lead to a decrease in other resident species, but this is not really quantified by the authors, although methods for it exist (see below). Similarly, strong shifts in species dominating the community but with similar evenness values might occur and hide real changes in dominance/evenness, and could be captured by shifts in rank-abundance curves.

-Hillebrand H, Blasius B, Borer ET, Chase JM, Downing JA, Eriksson BK, et al. Biodiversity change is uncoupled from species richness trends: Consequences for conservation and monitoring. *J Appl Ecol.* 2018;55(1):169-84.

-Avolio ML, Carroll IT, Collins SL, Houseman GR, Hallett LM, Isbell F, et al. A comprehensive approach to analyzing community dynamics using rank abundance curves. *Ecosphere.* 2019 Oct;10(10).

- While the functional indices used clearly show the change in trait composition, these are based on relative abundance distribution in the community. The arrival of new species in the community will obviously change this distribution (as shown in Figure 4), but not necessarily the absolute abundance/biomass of the resident species. Is it so that in the Barents Sea the abundance/biomass of benthic specialist species has already decreased, or that the changes in trait distributions are only due to the increase in abundance of invading species?

- The authors seem to confound functional diversity and ecosystem functions throughout the manuscript, or at least the definition of ecosystem functioning was not clear to me. In the introduction and discussion, I read the term 'ecosystem function' as biomass production, benthopelagic coupling or other ecosystem processes/properties. However, in the methods, it is indicated that the authors 'analysed the temporal development of key ecosystem function metrics'. Some food for thought literature below:

-Jax, K. 2005. Function and "functioning" in ecology: what does it mean? *Oikos*, 111(3):641-648.

-Bellwood, D. R., R. P. Streit, S. J. Brandl, and S. B. Tebbett 2019. The meaning of the term 'function' in ecology: A coral reef perspective. *Functional Ecology*, 33(6):948-961.

- Taxonomic diversity metrics are quite sensitive to sampling, but authors do not seem to standardize them, despite some substantial differences in sampling effort through time (as shown in Figure S1). The number of species and abundance distribution will typically be influenced by the sampling effort across sites and through time. This is more described in the publications below, with some ways to correct for differences in sampling effort:

-Gotelli, N. J. and R. K. Colwell 2001. Quantifying biodiversity: procedures and pitfalls in the measurement and comparison of species richness. *Ecology Letters*, 4(4):379-391.

-Hsieh, T. C., K. H. Ma, and A. Chao 2016. iNEXT: an R package for rarefaction and extrapolation of species diversity (Hill numbers). *Methods in Ecology and Evolution*, 7(12):1451-1456.

- The authors seem to use some model to explain functional dispersion by the other diversity metrics. Is it a linear model? I looked in the methods and was unable to locate any detail on this. I do not understand the point of explaining a functional index by other functional indices.

Furthermore, it is never shown how the multiple diversity indices are correlated (variance inflation factors should also be investigated) in the study and I would not be surprised that they are, because species richness and functional richness are usually correlated, just as species richness and evenness are not independent (Jost, 2010). Similarly, functional dispersion is correlated to species richness (see simulation study in Laliberté & Legendre, 2010, already cited by the authors), so I don't see how it explains anything.

-Jost, L. 2010. The Relation between Evenness and Diversity. *Diversity*, 2(2):207-232.

- I think it's a shame that the authors do not study changes in functional diversity indices for other areas than the two squared areas shown in Figure 1. Although the other areas might not go through as much changes (are they?), they can still provide interesting insights in the ecosystem.

- Functional indices are sensitive to how many traits are included, and may therefore change the temporal patterns observed by the authors. I think a better description of why the authors chose the traits they use, how correlated they are and what is the reasoning for choosing these traits should be included.

-Lefcheck JS, Bastazini V a. G, Griffin JN. Choosing and using multiple traits in functional diversity research. *Environ Conserv*. 2015 Jun;42(2):104-7

Review form: Reviewer 2

Recommendation

Major revision is needed (please make suggestions in comments)

Scientific importance: Is the manuscript an original and important contribution to its field?

Good

General interest: Is the paper of sufficient general interest?

Acceptable

Quality of the paper: Is the overall quality of the paper suitable?

Acceptable

Is the length of the paper justified?

No

Should the paper be seen by a specialist statistical reviewer?

No

Do you have any concerns about statistical analyses in this paper? If so, please specify them explicitly in your report.

Yes

It is a condition of publication that authors make their supporting data, code and materials available - either as supplementary material or hosted in an external repository. Please rate, if applicable, the supporting data on the following criteria.

Is it accessible?

No

Is it clear?

No

Is it adequate?

Yes

Do you have any ethical concerns with this paper?

No

Comments to the Author

Please see the attached file "Comments to Authors". (See Appendix A)

Decision letter (RSPB-2020-1458.R0)

14-Jul-2020

Dear Dr Frainer:

I am writing to inform you that your manuscript RSPB-2020-1458 entitled "Increased functional diversity warns of ecological transition in the Arctic" has, in its current form, been rejected for publication in Proceedings B.

This action has been taken on the advice of referees, who have recommended that substantial revisions are necessary. With this in mind we would be happy to consider a resubmission, provided the comments of the referees are fully addressed. However please note that this is not a provisional acceptance.

Sincerely,

Dr Daniel Costa

Associate Editor
 Board Member: 1
 Comments to Author:
 Dear Authors

The two expert reviewers found your article well written and interesting, but also came up with criticisms and suggestions how the ms could be still improved. One of the referees was in opinion that the novelty value in the ms is limited because the results and conclusions are too similar to those of the previous PNAS study by the lead author. Given this and the expert reviewer's recommendations (*viz.* Reject and Major revision), it would be hard for me to recommend anything but rejection. Nevertheless, given that reviews are always somewhat subjective and both reviewers commended many aspects of your study, I would like recommend rejection with possibility to resubmit. This would mean that you would have opportunity to respond to referee's criticism and take account their constructive comments. The ms would likely go back to same reviewers (plus an additional one) - hence, it would be in your discretion to weigh your options if you would like to resubmit to Proc B or elsewhere.

Best wishes,
 Juha Merilä

Reviewer(s)' Comments to Author:
 Referee: 1

Comments to the Author(s)

I have reviewed the manuscript entitled "Increased functional diversity warns of ecological transition in the Arctic" for Proceedings of the Royal Society B. This study is providing evidence for temporal changes in several taxonomic and functional diversity indices in the Barents Sea using scientific surveys and a collection of functional traits. This manuscript is concise and well-written, and it was a pleasure to read the study. This work seems to be a follow up study to a previous publication from 2017 entitled "Climate-driven changes in functional biogeography of Arctic marine fish communities", published in PNAS, and led by the same first author. Although the authors apply multiple diversity indices (functional dispersion, functional richness, species richness and evenness, community-weighted variance), I don't think the overall conclusions and findings are novel compared to the previous study published in 2017. The results from the study in 2017 are essentially shown in Figure 4a. Using community-weighted variance and functional dispersion seem to show similar results (Figure 3a/4b and Figure 2a, respectively). On the contrary to what the authors state (in the abstract and introduction), I don't find the methods employed to assess functional diversity that novel, as they have been developed more than a decade ago (and even more for the taxonomic indices used) and have been very frequently applied in ecology. However, even if not novel, it can of course provide interesting insights in community changes. I further have some comments on the methodology applied in the manuscript. I think the work done is of good quality, but I think the authors could dig in more in their analyses to better understand taxonomic and functional biodiversity dynamics with more insightful indices than richness and evenness. The ecosystem studied by the authors is very well documented and provides very important insights for management and conservation of species under climate change, so I encourage the authors to improve their analyses to further unravel diversity changes in the Barents Sea. I list below a number of points which I hope will help the authors to improve their study (I have not authored any of the literature sources mentioned below):

- The authors mention multiple times that changes in functional diversity will affect ecosystem functioning, and I think the authors could link changes in trait composition/ diversity to ecosystem functioning, by comparing with spatio-temporal changes in biomass/benthic-pelagic coupling through time. It has been evidenced that climate change is, through an increase in diversity and changes in species habitat use, increasing ecosystem functioning (for instance,

Friedland et al., 2020). It would be interesting to see if similar changes occur in the Barents Sea, and they can be attributed to changes in trait composition and taxonomic diversity.

-Friedland, K.D., Langan, J.A., Large, S.I., Selden, R.L., Link, J.S., Watson, R.A., Collie, J.S., 2020. Changes in higher trophic level productivity, diversity and niche space in a rapidly warming continental shelf ecosystem. *Science of the Total Environment* 704, 135270.

- The case of the Barents Sea is very interesting and the ecosystem is obviously going through major climate-driven changes. However, the trends in taxonomic and functional diversity found by the authors may hide more complex changes. Species richness is a well-known poor indicator to quantify complex biodiversity changes in communities in time and space. While the polar part of the ecosystem is inhabited by new species previously absent, each sub-system might still go through gains and losses of species. It would be interesting to apply methods providing evidence for gains in species and abundance of boreal species, as well as losses/decrease in benthic resident polar species. The authors particularly state that the increase in functional diversity is not beneficial for the ecosystem, but they do not show how un-beneficial it is in terms of local diversity. This is however well done for trait distributions in Figure 4, but I think the authors could go further in their analyses beyond trait distributions. I understand that the arrival of new species could lead to a decrease in other resident species, but this is not really quantified by the authors, although methods for it exist (see below). Similarly, strong shifts in species dominating the community but with similar evenness values might occur and hide real changes in dominance/evenness, and could be captured by shifts in rank-abundance curves.

-Hillebrand H, Blasius B, Borer ET, Chase JM, Downing JA, Eriksson BK, et al. Biodiversity change is uncoupled from species richness trends: Consequences for conservation and monitoring. *J Appl Ecol.* 2018;55(1):169–84.

-Avolio ML, Carroll IT, Collins SL, Houseman GR, Hallett LM, Isbell F, et al. A comprehensive approach to analyzing community dynamics using rank abundance curves. *Ecosphere.* 2019 Oct;10(10).

- While the functional indices used clearly show the change in trait composition, these are based on relative abundance distribution in the community. The arrival of new species in the community will obviously change this distribution (as shown in Figure 4), but not necessarily the absolute abundance/biomass of the resident species. Is it so that in the Barents Sea the abundance/biomass of benthic specialist species has already decreased, or that the changes in trait distributions are only due to the increase in abundance of invading species?

- The authors seem to confound functional diversity and ecosystem functions throughout the manuscript, or at least the definition of ecosystem functioning was not clear to me. In the introduction and discussion, I read the term 'ecosystem function' as biomass production, benthopelagic coupling or other ecosystem processes/properties. However, in the methods, it is indicated that the authors 'analysed the temporal development of key ecosystem function metrics'. Some food for thought literature below:

-Jax, K. 2005. Function and "functioning" in ecology: what does it mean? *Oikos*, 111(3):641–648.

-Bellwood, D. R., R. P. Streit, S. J. Brandl, and S. B. Tebbett 2019. The meaning of the term 'function' in ecology: A coral reef perspective. *Functional Ecology*, 33(6):948–961.

- Taxonomic diversity metrics are quite sensitive to sampling, but authors do not seem to standardize them, despite some substantial differences in sampling effort through time (as shown in Figure S1). The number of species and abundance distribution will typically be influenced by the sampling effort across sites and through time. This is more described in the publications below, with some ways to correct for differences in sampling effort:

-Gotelli, N. J. and R. K. Colwell 2001. Quantifying biodiversity: procedures and pitfalls in the measurement and comparison of species richness. *Ecology Letters*, 4(4):379–391.

-Hsieh, T. C., K. H. Ma, and A. Chao 2016. iNEXT: an R package for rarefaction and extrapolation of species diversity (Hill numbers). *Methods in Ecology and Evolution*, 7(12):1451–1456.

- The authors seem to use some model to explain functional dispersion by the other diversity metrics. Is it a linear model? I looked in the methods and was unable to locate any detail on this. I do not understand the point of explaining a functional index by other functional indices. Furthermore, it is never shown how the multiple diversity indices are correlated (variance inflation factors should also be investigated) in the study and I would not be surprised that they are, because species richness and functional richness are usually correlated, just as species richness and evenness are not independent (Jost, 2010). Similarly, functional dispersion is correlated to species richness (see simulation study in Laliberté & Legendre, 2010, already cited by the authors), so I don't see how it explains anything.

-Jost, L. 2010. The Relation between Evenness and Diversity. *Diversity*, 2(2):207–232.

- I think it's a shame that the authors do not study changes in functional diversity indices for other areas than the two squared areas shown in Figure 1. Although the other areas might not go through as much changes (are they?), they can still provide interesting insights in the ecosystem.

- Functional indices are sensitive to how many traits are included, and may therefore change the temporal patterns observed by the authors. I think a better description of why the authors chose the traits they use, how correlated they are and what is the reasoning for choosing these traits should be included.

-Lefcheck JS, Bastazini V a. G, Griffin JN. Choosing and using multiple traits in functional diversity research. *Environ Conserv.* 2015 Jun;42(2):104–7

Referee: 2

Comments to the Author(s)

Please see the attached file "Comments to Authors".

Author's Response to Decision Letter for (RSPB-2020-1458.R0)

See Appendix B.

RSPB-2021-0054.R0

Review form: Reviewer 1

Recommendation

Major revision is needed (please make suggestions in comments)

Scientific importance: Is the manuscript an original and important contribution to its field?

Good

General interest: Is the paper of sufficient general interest?

Excellent

Quality of the paper: Is the overall quality of the paper suitable?

Excellent

Is the length of the paper justified?

Yes

Should the paper be seen by a specialist statistical reviewer?

No

Do you have any concerns about statistical analyses in this paper? If so, please specify them explicitly in your report.

Yes

It is a condition of publication that authors make their supporting data, code and materials available - either as supplementary material or hosted in an external repository. Please rate, if applicable, the supporting data on the following criteria.

Is it accessible?

Yes

Is it clear?

Yes

Is it adequate?

Yes

Do you have any ethical concerns with this paper?

No

Comments to the Author

I think the manuscript has improved, especially regarding the novelty of the study/methods. I understand better the advantage of using the “functional variance” metric, and this elevates the originality of the manuscript. The authors responded well to most of the comments and included some new analyses in supplementary material providing more explanations of the changes occurring in the boreal and arctic regions of the Barents Sea. I am recommending major revision of the manuscript because I still have some misunderstandings about some parts, and some points the authors should work on to finalize their study (developed below). I want to point out that the main part of the manuscript focusing on the CWM/CWV is very clear for me and my remarks focus on the smaller analysis on the functional dispersion calculation and modeling. Authors can find minor comments at the end.

Sampling effort, biodiversity indices & linear model – I am not sure I understand how the taxonomic and functional diversity indices have been calculated. The authors mention the indices “rely on station-level estimates across all years” in their response, and “in each sampling location” L94 in the manuscript. Is each diversity index calculated at the haul level, for each year? Or are hauls somehow aggregated to some station/location/site level? If calculated at the haul level, there is indeed no sampling bias (at least not from the number of hauls) to take into account.

If calculated at the haul level, I think the subsequent model should be ran using all hauls for all years? I am not sure this was done by the authors, because Figure 2 shows aggregated averages. The authors mention they use averages from sub-regions avoiding a spatial autocorrelation issue (response to reviewer #2). Please be clear on what was done and how: (i) if the model uses aggregates of samples, indices should be corrected for sampling bias using rarefaction curves (ii) if at the haul level, the model has to include spatio-temporal autocorrelation, as mentioned by reviewer #2 (iii) if an analysis is done at a specific scale, it has to be clearly specified in the methods.

If metrics are averaged at the region level before running the linear model, this a bit strange from a statistical point of view. I would have at least preferred something similar to the following study (though diversity metrics not corrected for sampling bias):

-Greenstreet, S. P. R., Fraser, H. M., Rogers, S. I., Trenkel, V. M., Simpson, S. D., and Pinnegar, J. K. 2012. Redundancy in metrics describing the composition, structure, and functioning of the North Sea demersal fish community. – *ICES Journal of Marine Science*, 69: 8–22.

–Linear model –– In response to the authors’ comments, I don’t think that this linear model is proving any causation (do they ever?). I went through the –Laliberte ɪmp; Legendre 2010– publication: although I do understand how species richness, evenness and functional richness may be linked to functional dispersion, I don’t think the 2010 cited paper is particularly inviting to create a model where indices are by construction correlated (since that is what is shown in their simulation study). If authors see a clear benefit to model functional dispersion by other functional diversity indices, they should make sure to justify their choice further. Additionally, if the point is to disentangle the strongest link with richness/evenness/functional richness and functional dispersion (as explained in the authors’ response), it needs to be shown in the manuscript. So far, only the coefficient of determination is mentioned, and I could not find any estimate of the explanatory variable effects. Again, these explanatory variables are correlated, and I think that this needs to be evaluated before running the model (variance inflation factor, as mentioned in the previous review).

L145-147: “In the Arctic, functional richness, species richness and evenness increased faster than in the boreal region but their rate of increase does not explain the higher rate of change in functional dispersion observed in the Arctic (Fig. 2a-d).”

1. Yes, it seems so, and it’s a very interesting pattern – but it is not based on any statistical comparison, and could at least include some confidence intervals on the graphs?
2. The ‘effects’ of evenness/richness/functional richness hypothesized by the authors on the dispersion are not evident from the graphs and could be backed up with the model results?

–Comparing functional diversity indices –– I think it is clear from the analyses conducted that the most insightful one is the CWM-CWV analyses, which also represent the core of the paper. This could be more emphasized in the discussion, giving a perspective on which functional diversity metrics are relevant for studying composition change in communities.

–Small comments–

L39: typo, should be “evenness”

L40-43: something is wrong with that sentence: “our novel assessment of functional variance indicates that this trend might be short-lived”?

L89-92: “The chosen traits provide information on species characteristics that are involved in assembly processes (habitat affinity and feeding ecology), and ecosystem functions (body size, feeding ecology, and foodweb characteristics) [14].”

Here, it should not be mentioned as ecosystem functions – body size is not a function

L103-106: “To disentangle the relative contributions of functional richness, species evenness, and species richness on functional dispersion, we used a general linear model using data from the entire Barents Sea across all years.”

Here again, it’s not really clear what was done, I think more information needs to be detailed on how the data were treated. Is it a model per haul?

L143-144: “Together, these variables accounted for 44% of the spatio-temporal variation in functional dispersion across the Barents Sea (Fig. S3)”

All explanatory variables need to be mentioned here again for clarity (or in the previous sentence).

L144 “spatio-temporal variation”: if the model is based on the 2 regions, it’s not really spatio-temporal variation? Reading response to reviewer #2, it’s based on the aggregated regions?

L154: remove one “and”

L196: typo

Figure S3: please correct typos (“eveness”, “dispesion”) – why is functional richness also called dissimilarity? Dissimilarity measures are usually not richness of a community.

Review form: Reviewer 2 (Anna Törnroos)

Recommendation

Accept as is

Scientific importance: Is the manuscript an original and important contribution to its field?

Good

General interest: Is the paper of sufficient general interest?

Good

Quality of the paper: Is the overall quality of the paper suitable?

Good

Is the length of the paper justified?

Yes

Should the paper be seen by a specialist statistical reviewer?

No

Do you have any concerns about statistical analyses in this paper? If so, please specify them explicitly in your report.

No

It is a condition of publication that authors make their supporting data, code and materials available - either as supplementary material or hosted in an external repository. Please rate, if applicable, the supporting data on the following criteria.

Is it accessible?

Yes

Is it clear?

Yes

Is it adequate?

Yes

Do you have any ethical concerns with this paper?

No

Comments to the Author

In my opinion the Authors have adressed the Reviewers' comments adequately.

Decision letter (RSPB-2021-0054.R0)

15-Feb-2021

Dear Dr Frainer:

Your manuscript has now been peer reviewed and the reviews have been assessed by an Associate Editor. The reviewers' comments (not including confidential comments to the Editor) and the comments from the Associate Editor are included at the end of this email for your reference. As you will see, the reviewers and the Editors have raised some concerns with your manuscript and we would like to invite you to revise your manuscript to address them.

Research ethics:

Use of animals and field studies:

It is a condition of publication that you make available the data and research materials supporting the results in the article (<https://royalsociety.org/journals/authors/author-guidelines/#data>). Datasets should be deposited in an appropriate publicly available repository and details of the associated accession number, link or DOI to the datasets must be included in the Data Accessibility section of the article (<https://royalsociety.org/journals/ethics->

policies/data-sharing-mining/). Reference(s) to datasets should also be included in the reference list of the article with DOIs (where available).

Please submit a copy of your revised paper within three weeks. If we do not hear from you within this time your manuscript will be rejected. If you are unable to meet this deadline please let us know as soon as possible, as we may be able to grant a short extension.

Best wishes,
Dr Daniel Costa
mailto: proceedingsb@royalsociety.org

Associate Editor Board Member

Comments to Author:

Dear Authors,

Thank you for revising your ms - as you can see from the reviewer comments, both referees think the ms has improved considerably. However, the other referee has still some very specific suggestions how you could improve the ms, and I would like to ask you to consider these. This would make your work more accessible and increase its value I believe.

Best wishes,
Juha M

Reviewer(s)' Comments to Author:

Referee: 1

Comments to the Author(s).

I think the manuscript has improved, especially regarding the novelty of the study/methods. I understand better the advantage of using the “functional variance” metric, and this elevates the originality of the manuscript. The authors responded well to most of the comments and included some new analyses in supplementary material providing more explanations of the changes occurring in the boreal and arctic regions of the Barents Sea. I am recommending major revision of the manuscript because I still have some misunderstandings about some parts, and some points the authors should work on to finalize their study (developed below). I want to point out that the main part of the manuscript focusing on the CWM/CWV is very clear for me and my remarks focus on the smaller analysis on the functional dispersion calculation and modeling. Authors can find minor comments at the end.

Sampling effort, biodiversity indices & linear model – I am not sure I understand how the taxonomic and functional diversity indices have been calculated. The authors mention the indices “rely on station-level estimates across all years” in their response, and “in each sampling location” L94 in the manuscript. Is each diversity index calculated at the haul level, for each year? Or are hauls somehow aggregated to some station/location/site level? If calculated at the haul level, there is indeed no sampling bias (at least not from the number of hauls) to take into account.

If calculated at the haul level, I think the subsequent model should be ran using all hauls for all years? I am not sure this was done by the authors, because Figure 2 shows aggregated averages. The authors mention they use averages from sub-regions avoiding a spatial autocorrelation issue (response to reviewer #2). Please be clear on what was done and how: (i) if the model uses aggregates of samples, indices should be corrected for sampling bias using rarefaction curves (ii) if at the haul level, the model has to include spatio-temporal autocorrelation, as mentioned by reviewer #2 (iii) if an analysis is done at a specific scale, it has to be clearly specified in the methods.

If metrics are averaged at the region level before running the linear model, this a bit strange from a statistical point of view. I would have at least preferred something similar to the following study (though diversity metrics not corrected for sampling bias):

-Greenstreet, S. P. R., Fraser, H. M., Rogers, S. I., Trenkel, V. M., Simpson, S. D., and Pinnegar, J. K. 2012. Redundancy in metrics describing the composition, structure, and functioning of the North Sea demersal fish community. – *ICES Journal of Marine Science*, 69: 8–22.

Linear model – In response to the authors’ comments, I don’t think that this linear model is proving any causation (do they ever?). I went through the Laliberte & Legendre 2010 publication: although I do understand how species richness, evenness and functional richness may be linked to functional dispersion, I don’t think the 2010 cited paper is particularly inviting to create a model where indices are by construction correlated (since that is what is shown in their simulation study). If authors see a clear benefit to model functional dispersion by other functional diversity indices, they should make sure to justify their choice further. Additionally, if the point is to disentangle the strongest link with richness/evenness/functional richness and functional dispersion (as explained in the authors’ response), it needs to be shown it in the manuscript. So far, only the coefficient of determination is mentioned, and I could not find any estimate of the explanatory variable effects. Again, these explanatory variables are correlated, and I think that this needs to be evaluated before running the model (variance inflation factor, as mentioned in the previous review).

L145-147: “In the Arctic, functional richness, species richness and evenness increased faster than in the boreal region but their rate of increase does not explain the higher rate of change in functional dispersion observed in the Arctic (Fig. 2a-d).”

1. Yes, it seems so, and it's a very interesting pattern – but it is not based on any statistical comparison, and could at least include some confidence intervals on the graphs?
2. The 'effects' of evenness/richness/functional richness hypothesized by the authors on the dispersion are not evident from the graphs and could be backed up with the model results?

Comparing functional diversity indices – I think it is clear from the analyses conducted that the most insightful one is the CWM-CWV analyses, which also represent the core of the paper. This could be more emphasized in the discussion, giving a perspective on which functional diversity metrics are relevant for studying composition change in communities.

Small comments

L39: typo, should be “evenness”

L40-43: something is wrong with that sentence: “our novel assessment of functional variance indicates that this trend might be short-lived”?

L89-92: “The chosen traits provide information on species characteristics that are involved in assembly processes (habitat affinity and feeding ecology), and ecosystem functions (body size, feeding ecology, and foodweb characteristics) [14].”

Here, it should not be mentioned as ecosystem functions – body size is not a function

L103-106: “To disentangle the relative contributions of functional richness, species evenness, and species richness on functional dispersion, we used a general linear model using data from the entire Barents Sea across all years.”

Here again, it's not really clear what was done, I think more information needs to be detailed on how the data were treated. Is it a model per haul?

L143-144: “Together, these variables accounted for 44% of the spatio-temporal variation in functional dispersion across the Barents Sea (Fig. S3)”

All explanatory variables need to be mentioned here again for clarity (or in the previous sentence).

L144 “spatio-temporal variation”: if the model is based on the 2 regions, it's not really spatio-temporal variation? Reading response to reviewer #2, it's based on the aggregated regions?

L154: remove one “and”

L196: typo

Figure S3: please correct typos (“evenness”, “dispesion”) – why is functional richness also called dissimilarity? Dissimilarity measures are usually not richness of a community.

Referee: 2

Comments to the Author(s).

In my opinion the Authors have addressed the Reviewers' comments adequately.

Author's Response to Decision Letter for (RSPB-2021-0054.R0)

See Appendix C.

Decision letter (RSPB-2021-0054.R1)

10-Mar-2021

Dear Dr Frainer

I am pleased to inform you that your manuscript entitled "Increased functional diversity warns of ecological transition in the Arctic" has been accepted for publication in Proceedings B.

Data Accessibility section

Open Access

Your article has been estimated as being 6 pages long. Our Production Office will be able to confirm the exact length at proof stage.

Paper charges

Sincerely,

Dr Daniel Costa

Associate Editor:
Board Member
Comments to Author:
Dear Authors,

Thank you for revising the ms along the lines suggested by the reviewer - I believe you have done good job and I have no further comments.

Best

Juha

Appendix A

RSPB-2020-1458

Comments to Authors

The study presents an assessment of the development of functional diversity (richness, dispersion, community-weighted mean and variance) of the Barents Sea fish community in space (particularly focused on a boreal and an Arctic area) and over time (from 2012-2017). It includes 51 fish species and trawl data (shrimp bottom trawls) from 4223 samples, and 16 traits totalling 24 individual traits. By multivariate, and I suppose, based on the graphic presentations, some univariate analysis the changes in functional measures are assessed. Results show that especially functional dispersion has changed in a fast pace in the Arctic region of the study. This increase is further assessed and attributed partly to increases in species and their evenness, but also to the community trait variance and incoming species having other traits than the Arctic resident ones.

The study is scientifically good and presents an interest to a wider audience. The language is good and figures in general good and interesting (especially Fig. 4). However, in my opinion, there are several issues that needs to be addressed in order for the manuscript to potentially be accepted. Most of my comments regard the at times weak link between what is assessed and what is presented. For example, supplementary material is not referred to in the text, which I believe would make the argumentation and presentation of results clearer. Also, since the traits are not presented here, rather referenced to another paper, but still discussed on both general and more detailed level, as well as argued to explain parts of the results, it is difficult for the reader to follow. Please see my detailed comments on these and other issues below:

Specific comments:

Abstract

- I believe the Abstract would benefit from focusing more on the biodiversity changes, which are here assessed, rather than mentioning of functions. I also believe that the statements in the end by referring to traits per se is not reflecting the results presented in the manuscript, rather they are arguments in a discussion. If the case of what traits are influencing . In essence, please re-write the Abstract so that it reflects the analysis and results better.

Materials and Methods

- p. 5. You mention 51 fish species here in the text, and 52 species in the Supplementary material, please check and clarify.
- p. 6. Sentence starting: "Together, the CWV and... "
- can you in this or the following sentence state that you use the PC1 for this analysis, or clarify here (perhaps with keeping the supplementary material in mind).
 - What about autocorrelation, was this checked?
 - What are the pre-defined regions based on?
 - What about sampling size (# between years eg. 2014 and 2017), was this taken into consideration?

Results and Discussion:

p.6 – The reference to Fig. 2b-d. I don't understand the sentence in respect to the figure, do you perhaps mean Supplementary figure S2?

p7. – The reference to "...57% of variation in the data...". Would this again be reference to supplementary material? In addition, in Figure S3, this % is 56%, why the inconsistency? In addition, where does the correlation come from, please clarify the linkage between results and figures.

p.7 – The reference to foraging-related traits associated with PCI. Can you name or clarify what these traits are, and what they are called in your analysis e.g. in the supplementary figures?!

p.7 – The sentence and paragraph starting "The strong increase in functional diversity" . What do you mean here (dispersion)?! I would suggest checking your usage of the term diversity in reference to what you want to

highlight for the reader, sometimes it is used broadly in a fair way, but sometimes it is used when one of the specific metrics could rather be spelled out. Also, check, throughout this section, that you back up your statements with either your own results and reference them properly, and/or previous studies. This comment refers to this and many of the sentences and arguments after this.

p.8 – The mentioning of “habitat modifications” in the second paragraph. Is it really habitat modifications, or perhaps biotope or niche you mean?!

p.8 – In the same sentence in which the habitat modification is mentioned, you refer to the benthivore fish component being small, although you argue for (and this is shown in many other studies) that this component is very large in the Arctic. I would remove the word small.

p.8. The last sentence in the second paragraph is really well written and important. This is something for future studies to pick up.

Figures

- In Figure 1 the areas indicated are not described! Also, you refer to these in figure 2, so check the link between these figures again. In addition, check labelling of A, B, C etc. these are missing.
- Figure 1. I would also argue that the figure title is a bit grandiose as you do not link to any environmental changes in the figures or analysis.
- Figure 2. What are these boxes you refer to? Clarify.
- Figure 2. How are these smoothed average calculated, and why not moving averages?
- Figure 3. In a you state that you present “Functional diversity”, but this would in my opinion correctly be called functional mean variance here, not functional diversity. In addition, I would explicitly specify that this is based on PC1 in the y-axis legend.
- Figure 3. Change b, d to b-d.
- Figure 3. You present the developments of two traits, but what are these traits really? What is development rate, please briefly explain, e.g. in brackets in the legend.
- Figure 4. Although I like the presentation of the results in this way, I would provide some more info in the figure legend, eg. add that is shows the development of the period from 2004 to 2017, based on areas or all stations/trawls? and is it PC1?

Appendix B

Dear Editor,

We are grateful for the opportunity to resubmit our paper ‘Increased functional diversity warns of ecological transition in the Arctic’, which we have revised and improved with the help of the constructive comments and suggestions from the reviewers. During revision, we have carefully addressed all reviewers’ comments, which we answer in our detailed response to reviewers below.

We would like to highlight that, to answer reviewer #1 comments, we have emphasised the novelty of our findings and conclusions, distinguishing the scientific contribution of our manuscript on functional diversity from that of a previous publication by our group dealing with functional characterization. Further, we have clarified which are the new elements of our approach to functional diversity estimation based on multivariate analysis of community-weighted traits variance. By allowing to identify the traits contributing to change in functional diversity, our functional variance approach is instrumental for our inferences and conclusions on community assembly processes during Arctic colonization by boreal species and on adaptive capacity of ecosystems.

Sincerely,

André

Detailed response to reviewers:

- Referee: 1

I have reviewed the manuscript entitled “Increased functional diversity warns of ecological transition in the Arctic” for Proceedings of the Royal Society B. This study is providing evidence for temporal changes in several taxonomic and functional diversity indices in the Barents Sea using scientific surveys and a collection of functional traits. This manuscript is concise and well-written, and it was a pleasure to read the study.

**Our response: Thank you very much for your positive remarks.*

This work seems to be a follow up study to a previous publication from 2017 entitled “Climate-driven changes in functional biogeography of Arctic marine fish communities”, published in PNAS, and led by the same first author.

Although the authors apply multiple diversity indices (functional dispersion, functional richness, species richness and evenness, community-weighted variance), I don’t think the overall conclusions and findings are novel compared to the previous study published in 2017. The results from the study in 2017 are essentially shown in Figure 4a. Using community-weighted variance and functional dispersion seem to show similar results (Figure 3a/4b and Figure 2a, respectively).

**Our response: The current manuscript is a new scientific contribution which differs from our 2017 paper (Frainer et al., PNAS) with regard to questions, main findings and conclusions. In our manuscript, we have harmonized elements of the approach (e.g. geographical focus, target species, functional traits data) and graphical summaries with our previous studies to facilitate comparison and integration. But the 2017 paper dealt with the functional characterization of fish communities, i.e. what are the typical traits of fish in a given region, and showed a climate driven shift in the *mean* values of the main traits characterizing fish communities from 2004 to 2012. In the current manuscript, we address the implications of climate change for fish biodiversity, focusing on functional diversity, i.e. on how and why the traits of fish differ in a given region, and our main finding shows climate driven change in the *spread* of functional traits related to foraging. In our current manuscript, Figure 4a is the only result that is an extension of the 2017 paper, adding five years to the climate driven trend in fish functional characterization based on community-weighted mean trait values. Figure 4a is emphatically not a main result of our new manuscript, but complements the novel findings on functional diversity to support*

our discussion and inferences. Figure 4a is based on measures of central tendency (community weighted means) in functional traits, not on functional traits spread or diversity shown in figures 2a, 3a or 4b; figure 4a can thereby not show the same results as the other figures listed above. In an earlier draft, figure 4a was placed in the appendix and it may have to be moved back there if it distracts the reader from the main focus and findings of the study. With regard to novelty of findings and conclusions, we here address functional diversity as a mean to infer the assembly process during Arctic colonization by boreal species and to assess implications for adaptive capacity of ecosystems, none of which can be addressed by the *mean* functional traits values presented in figure 4a or in the 2017 PNAS paper. To address our objectives we introduce a novel approach to functional diversity estimation relying on multivariate functional traits variance analysis (please, see below our response to the specific comment regarding these analyses). The resulting metric, which we now refer to as *functional variance* following a suggestion by reviewer #2, allows to identify the traits contributing to functional diversity and thereby expands the scope of inferences that can be drawn from functional diversity analyses.

On the contrary to what the authors state (in the abstract and introduction), I don't find the methods employed to assess functional diversity that novel, as they have been developed more than a decade ago (and even more for the taxonomic indices used) and have been very frequently applied in ecology. However, even if not novel, it can of course provide interesting insights in community changes.

*Our response: Community-weighted *variance* has indeed been developed and suggested as early as last decade, as the reviewer points out; in the methods section we cite Enquist et al. 2015 in this respect. The novelty of our approach lies in the use of multiple traits, and the multivariate analysis of their variances (as now more explicitly stated in the methods section, L111-123), which allows to identify which traits contribute to functional diversity in a community. Nonetheless, following the comment from the reviewer, we have opted for tuning down our statement in L71, and instead mention the approach more straightforwardly. L71 now reads: 'a community-wide multi-trait analysis'.

I further have some comments on the methodology applied in the manuscript. I think the work done is of good quality, but I think the authors could dig in more in their analyses to better understand taxonomic and functional biodiversity dynamics with more insightful indices than richness and evenness. The ecosystem studied by the authors is very well documented and provides very important insights for management and conservation of species under climate change, so I encourage the authors to improve their analyses to further unravel diversity changes in the Barents Sea. I list below a number of points which I hope will help the authors to improve their study (I have not authored any of the literature sources mentioned below):

*Our response: Thank you for the suggestions and for the related literature, which helped us improve the manuscript.

- The authors mention multiple times that changes in functional diversity will affect ecosystem functioning, and I think the authors could link changes in trait composition/diversity to ecosystem functioning, by comparing with spatio-temporal changes in biomass/benthic-pelagic coupling through time. It has been evidenced that climate change is, through an increase in diversity and changes in species habitat use, increasing ecosystem functioning (for instance, Friedland et al., 2020). It would be interesting to see if similar changes occur in the Barents Sea, and they can be attributed to changes in trait composition and taxonomic diversity.

-Friedland, K.D., Langan, J.A., Large, S.I., Selden, R.L., Link, J.S., Watson, R.A., Collie, J.S., 2020. Changes in higher trophic level productivity, diversity and niche space in a rapidly warming continental shelf ecosystem. *Science of the Total Environment* 704, 135270.

*Our response: We considered this possibility during our revision work, but unfortunately, the available data on pelagic fish and other compartments of the Barents Sea ecosystem do not have a comparable level of resolution or quality as for the demersal component. Our colleagues are currently working on a massive task to quality-control and standardize data for the benthic compartment, which

includes more than 3000 species, but this is far from being ready for comparative analyses. Data on pelagic fish is even more precarious due to unstandardized samplings and a need for further quality control on the available data. Thus, it is currently impossible, though it would certainly be of great interest, to have measures of ecosystem functioning or other measures from the pelagic compartment, as the reviewer suggested.

- The case of the Barents Sea is very interesting and the ecosystem is obviously going through major climate-driven changes. However, the trends in taxonomic and functional diversity found by the authors may hide more complex changes. Species richness is a well-known poor indicator to quantify complex biodiversity changes in communities in time and space. While the polar part of the ecosystem is inhabited by new species previously absent, each sub-system might still go through gains and losses of species. It would be interesting to apply methods providing evidence for gains in species and abundance of boreal species, as well as losses/decrease in benthic resident polar species. The authors particularly state that the increase in functional diversity is not beneficial for the ecosystem, but they do not show how un-beneficial it is in terms of local diversity. This is however well done for trait distributions in Figure 4, but I think the authors could go further in their analyses beyond trait distributions. I understand that the arrival of new species could lead to a decrease in other resident species, but this is not really quantified by the authors, although methods for it exist (see below). Similarly, strong shifts in species dominating the community but with similar evenness values might occur and hide real changes in dominance/evenness, and could be captured by shifts in rank-abundance curves.

-Hillebrand H, Blasius B, Borer ET, Chase JM, Downing JA, Eriksson BK, et al. Biodiversity change is uncoupled from species richness trends: Consequences for conservation and monitoring. *J Appl Ecol.* 2018;55(1):169–84.

-Avolio ML, Carroll IT, Collins SL, Houseman GR, Hallett LM, Isbell F, et al. A comprehensive approach to analyzing community dynamics using rank abundance curves. *Ecosphere.* 2019 Oct;10(10).

*Our response: Thank you very much for this suggestion, which we considered including in the manuscript results. However, a recent publication focusing on beta diversity in the Barents Sea has explored species turnover as suggested by the reviewer, we thereby cite and integrate these findings in our discussion, L189.

- While the functional indices used clearly show the change in trait composition, these are based on relative abundance distribution in the community. The arrival of new species in the community will obviously change this distribution (as shown in Figure 4), but not necessarily the absolute abundance/biomass of the resident species. Is it so that in the Barents Sea the abundance/biomass of benthic specialist species has already decreased, or that the changes in trait distributions are only due to the increase in abundance of invading species?

*Our response: Thank you for this comment and suggestion, which we address in the revised manuscript. We now provide three new figures in the appendix, which show that mean abundance in the Arctic has fluctuated during the study period (Fig. S6), mostly due to increases in abundance of several demersal fish species, both boreal and Arctic ones (Fig. S7, and S8), often concomitant with extreme environmental conditions (low sea ice cover and high water temperatures – 2006, 2012, 2016). The main reduction in Arctic fish abundance observed in the Barents Sea happens with ice-associated pelagic species, as Polar cod, which are not included in our analyses. Nonetheless, declining abundances are observed for some of the non-dominant Arctic fish species in our analyses, but due to their relatively low abundances they have a minor influence on our weighted analyses of functional diversity. We show these species-specific changes in abundance in Fig. S7.

- The authors seem to confound functional diversity and ecosystem functions throughout the manuscript, or at least the definition of ecosystem functioning was not clear to me. In the introduction and discussion, I read the term ‘ecosystem function’ as biomass production, benthic-pelagic coupling or other ecosystem processes/properties. However, in the methods, it is indicated that the authors

'analysed the temporal development of key ecosystem function metrics'. Some food for thought literature below:

-Jax, K. 2005. Function and "functioning" in ecology: what does it mean? *Oikos*, 111(3):641–648.

-Bellwood, D. R., R. P. Streit, S. J. Brandl, and S. B. Tebbett 2019. The meaning of the term 'function' in ecology: A coral reef perspective. *Functional Ecology*, 33(6):948–961.

*Our response: Thank you for pointing this out. Indeed, writing of ecosystem functions played by species and of ecosystem functioning can give rise to confusing statements. We have clarified the sentences addressing these issues and our definition of functioning. L125 now reads: "We analysed the temporal development of functional traits related to key ecosystem functions"

- Taxonomic diversity metrics are quite sensitive to sampling, but authors do not seem to standardize them, despite some substantial differences in sampling effort through time (as shown in Figure S1). The number of species and abundance distribution will typically be influenced by the sampling effort across sites and through time. This is more described in the publications below, with some ways to correct for differences in sampling effort:

-Gotelli, N. J. and R. K. Colwell 2001. Quantifying biodiversity: procedures and pitfalls in the measurement and comparison of species richness. *Ecology Letters*, 4(4):379–391.

-Hsieh, T. C., K. H. Ma, and A. Chao 2016. iNEXT: an R package for rarefaction and extrapolation of species diversity (Hill numbers). *Methods in Ecology and Evolution*, 7(12):1451–1456.

*Our response: The reviewer is right that sampling bias may affect diversity metrics. In the manuscript we do not apply inferential statistics on the region-specific taxonomic diversity estimates where sampling bias could be a problem. We do use taxonomic diversity estimates as predictors in the model of functional dispersion, but these rely on station-level estimates across all years, and are thereby not affected by sampling effort in specific regions. Regarding sampling effort within the two regions addressed in our work, the main problem is found in the Arctic region in 2014, when the sampling effort decreased from above 45 stations sampled yearly to only 26. This reduced sampling is caused by the unusually extensive sea ice cover, which limited the possibilities for trawl sampling in the region. There is no obvious anomaly in richness and evenness values in 2014 for the Arctic region (Fig. 2), but we now warn the reader in the methods section about the reduced sampling effort in that year.

- The authors seem to use some model to explain functional dispersion by the other diversity metrics. Is it a linear model? I looked in the methods and was unable to locate any detail on this. I do not understand the point of explaining a functional index by other functional indices. Furthermore, it is never shown how the multiple diversity indices are correlated (variance inflation factors should also be investigated) in the study and I would not be surprised that they are, because species richness and functional richness are usually correlated, just as species richness and evenness are not independent (Jost, 2010). Similarly, functional dispersion is correlated to species richness (see simulation study in Laliberté & Legendre, 2010, already cited by the authors), so I don't see how it explains anything.

-Jost, L. 2010. The Relation between Evenness and Diversity. *Diversity*, 2(2):207–232.

*Our response: The statistical model of functional dispersion was a general linear model, and we now describe it in L102-106 in the revised manuscript. The reason for modelling functional dispersion as function of functional richness and species richness and evenness was to try and disentangle what are the causal pathways that lead to higher dispersion, as pointed out by Laliberté & Legendre, 2010. Climate warming could affect functional dispersion via changes in species richness, evenness or functional richness (driven by changes in species composition), or a combination of those. We were interested in finding which was the most important factor driving the change in functional dispersion, and what were their relationships. The coefficient of determination for the model is updated to 0.44 because the model now includes the entire Barents Sea data and not just samples from the two separate regions.

- I think it's a shame that the authors do not study changes in functional diversity indices for other areas than the two squared areas shown in Figure 1. Although the other areas might not go through

as much changes (are they?), they can still provide interesting insights in the ecosystem.
*Our response: Indeed, the analyses on temporal changes in functional diversity focused on the two sub-regions that better characterize the two contrasting zoogeographic areas of the Barents Sea, boreal and Arctic. The other regions in the Barents Sea comprise areas of mixed waters (the polar front) or less well-defined zoogeography than the boreal and Arctic sub-regions included in our work. Nonetheless, we also provide information of change for all other areas in the Barents Sea, and these are included in the maps in Fig 1, and also in Supplementary Material Figures S1 and S2.

- Functional indices are sensitive to how many traits are included, and may therefore change the temporal patterns observed by the authors. I think a better description of why the authors chose the traits they use, how correlated they are and what is the reasoning for choosing these traits should be included.

-Lefcheck JS, Bastazini V a. G, Griffin JN. Choosing and using multiple traits in functional diversity research. *Environ Conserv.* 2015 Jun;42(2):104–7

*Our response: We agree with the reviewer that analyses using traits have to pay special attention to which and how many traits are included in it. Trait selection plays a pivotal role for the detection of effects in the target ecosystem and in providing a basis for inference. Also, some analyses may be biased by correlations between traits. The advantage of our multivariate approach when analyzing community-weighted variance is that the multivariate analysis allows for correlated traits to be analyzed together. With regard to rational for our choice of traits, in the methods section we refer to the literature and our previous 2017 paper (Frainer et al. PNAS), and in the revised manuscript we provide more detailed considerations in L89-92.

- Referee: 2

The study presents an assessment of the development of functional diversity (richness, dispersion, community-weighted mean and variance) of the Barents Sea fish community in space (particularly focused on a boreal and an Arctic area) and over time (from 2012-2017). It includes 51 fish species and trawl data (shrimp bottom trawls) from 4223 samples, and 16 traits totalling 24 individual traits. By multivariate, and I suppose, based on the graphic presentations, some univariate analysis the changes in functional measures are assessed. Results show that especially functional dispersion has changed in a fast pace in the Arctic region of the study. This increase is further assessed and attributed partly to increases in species and their evenness, but also to the community trait variance and incoming species having other traits than the Arctic resident ones. The study is scientifically good and presents an interest to a wider audience. The language is good and figures in general good and interesting (especially Fig. 4).

*Our response: Thank you very much for your positive remarks.

However, in my opinion, there are several issues that needs to be addressed in order for the manuscript to potentially be accepted. Most of my comments regard the at times weak link between what is assessed and what is presented. For example, supplementary material is not referred to in the text, which I believe would make the argumentation and presentation of results clearer. Also, since the traits are not presented here, rather referenced to another paper, but still discussed on both general and more detailed level, as well as argued to explain parts of the results, it is difficult for the reader to follow. Please see my detailed comments on these and other issues below:

*Our response: Thank you for your comments, which helped to improve the manuscript. As suggested, in the revised manuscript we have added references to the Supplementary Material throughout the text, making the argumentation and explanation of the results more clear.

Specific comments:

Abstract

- I believe the Abstract would benefit from focusing more on the biodiversity changes, which are here assessed, rather than mentioning of functions. I also believe that the statements in the end by

referring to traits per se is not reflecting the results presented in the manuscript, rather they are arguments in a discussion. If the case of what traits are influencing. In essence, please re-write the Abstract so that it reflects the analysis and results better.

*Our response: We have amended the abstract increasing the focus on biodiversity, as suggested. In the revised abstract, we added a key result on the increase in species richness and evenness, which was omitted in the original text. The description of traits at the end of the abstract reflects the outcome of our novel multivariate CWV analysis, which allows to identify the traits responsible for the change in functional diversity.

Materials and Methods

p. 5. You mention 51 fish species here in the text, and 52 species in the Supplementary material, please check and clarify.

*Our response: Thank you for noticing this typo. Our initial data set contained 51 species, but two species didn't have enough information on trait values. Since their abundances were also very small, not affecting our community-weighted analyses, we decide to leave them out. Thus, our manuscript has now been corrected and indicates 49 species.

p. 6. Sentence starting: "Together, the CWV and..."

- can you in this or the following sentence state that you use the PC1 for this analysis, or clarify here (perhaps with keeping the supplementary material in mind).

*Our response: In light of the reviewers comments and suggestions, we have now opted for renaming our functional metric "Functional variance". In the relevant sentence, we added (L119): "Hence, we name the values obtained from the first principal component as functional variance PC1, to distinguish them from the other components." Furthermore, we added a longer explanation of how the Functional variance is calculated in the same paragraph, which also answers to a comment from reviewer #1.

- What about autocorrelation, was this checked?

*Our response: Possible problems arising from spatial autocorrelation in the linear models were avoided by using the mean values from each sub-region mentioned in the methods.

- What are the pre-defined regions based on?

*Our response: We have now clarified what the pre-defined regions are, the revised text reads (L125-134):

"We analysed the temporal development of biodiversity and functional traits related to key ecosystem functions (see more explanation in the Results and Discussion section) in two distinct climatic and zoogeographic regions of the Barents Sea, a northeastern Arctic and a southwestern boreal region (Fig. 1). For taxonomic diversity indices, variation in sampling effort within regions may bias the estimates. The two regions have good sampling coverage during the study period with the exception of the low 2014 sampling effort in the Arctic region, caused by extensive sea-ice coverage limiting survey sampling. The regions have been analysed in previous publications for species distribution [2], food web metrics [22], and community-weighted mean values [1], providing important background ecological information for this study."

- What about sampling size (# between years eg. 2014 and 2017), was this taken into consideration?

*Our response: Please, see our response to a similar question raised by reviewer #1 on possible sampling bias. We address this issue now in the methods.

Results and Discussion:

p.6 – The reference to Fig. 2b-d. I don't understand the sentence in respect to the figure, do you perhaps mean Supplementary figure S2?

*Our response: The reference should have been to all plots of Figure 2, we have corrected this.

p7. – The reference to "...57% of variation in the data...". Would this again be reference to supplementary material?

In addition, in Figure S3, this % is 56%, why the inconsistency?

*Our response: Yes, thank you for noticing this. We added the reference to Fig. S4 (previously Fig. S3) and corrected the value in the text to 56%.

In addition, where does the correlation come from, please clarify the linkage between results and figures.

*Our response: We rephrased the discussion to more clearly describe where the correlation comes from: L152: “The functional variance PC1 was a good indicator of functional dispersion, being highly correlated with that metric ($r^2 = 0.83$).”

p.7 – The reference to foraging-related traits associated with PCI. Can you name or clarify what these traits are, and what they are called in your analysis e.g. in the supplementary figures?!

*Our response: We included a reference to Fig. S3 and added a complete trait table to the supplementary material, which clarifies which traits are the foraging-related ones.

p.7 – The sentence and paragraph starting “The strong increase in functional diversity”. What do you mean here (dispersion)?! I would suggest checking your usage of the term diversity in reference to what you want to highlight for the reader, sometimes it is used broadly in a fair way, but sometimes it is used when one of the specific metrics could rather be spelled out. Also, check, throughout this section, that you back up your statements with either your own results and reference them properly, and/or previous studies. This comment refers to this and many of the sentences and arguments after this.

*Our response: Thank you for pointing this out. We have corrected the text and added a more detailed description of our measure of functional diversity, which we chose to call ‘functional variance’, to distinguish it from other meanings of functional diversity, as pointed out by the reviewer. The text in L111 now reads:

“We then investigated the CWV of the 15 traits simultaneously by computing a principal component analysis (PCA). The principal components of the CWV PCA provide the basis for metrics of functional variance, allowing the simultaneous assessment of the magnitude (by means of the CWV scores on the principal components) and character (by inspecting the CWV factor loadings on the principal components) of functional variance. Thus, the functional variance approach proposed here provides a good candidate metric and can replace traditional indices of functional diversity (e.g., functional dispersion, functional richness, or functional evenness) lacking information on which functional traits are responsible for the functional diversity.”

p.8 – The mentioning of “habitat modifications” in the second paragraph. Is it really habitat modifications, or perhaps biotope or niche you mean?!

*Our response: We refer here to change in habitat characteristics caused by climate change, such as higher water temperatures or reduced ice cover.

p.8 – In the same sentence in which the habitat modification is mentioned, you refer to the benthivore fish component being small, although you argue for (and this is shown in many other studies) that this component is very large in the Arctic. I would remove the word small.

*Our response: Done

p.8. The last sentence in the second paragraph is really well written and important. This is something for future studies to pick up.

*Our response: Thank you!

Figures

- In Figure 1 the areas indicated are not described! Also, you refer to these in figure 2, so check the link between these figures again. In addition, check labelling of A, B, C etc. these are missing.

*Our response: Thank you for noticing this. We added a description of the areas in the methods (L125-134). The A-D labels were in the plots, but are positioned within the maps, and thus less visible.

- Figure 1. I would also argue that the figure title is a bit grandiose as you do not link to any environmental changes in the figures or analysis.

*Our response: We agree and have changed the figure title.

- Figure 2. What are these boxes you refer to? Clarify.

*Our response: Because of the changes in the Material and Methods, and Fig 1 legend, following the reviewer's comments, we have opted for changing this reference as well. It now reads: "polygons shown in Fig. 1)".

- Figure 2. How are these smoothed average calculated, and why not moving averages?

*Our response: We use loess estimations of conditional means, which does a weighted average estimation that downplays samples further away from the year of interest.

- Figure 3. In a you state that you present "Functional diversity", but this would in my opinion correctly be called functional mean variance here, not functional diversity. In addition, I would explicitly specify that this is based on PC1 in the y-axis legend.

*Our response: Thank you for the suggestion. We have renamed our index 'functional variance', which helps clarify some of the issues on terminology mentioned by the reviewer. Several parts of the text were changed to better clarify this issue, including the title.

- Figure 3. Change b, d to b-d.

*Our response: Done

- Figure 3. You present the developments of two traits, but what are these traits really? What is development rate, please briefly explain, e.g. in brackets in the legend.

*Our response: We corrected the term development rate to growth rate (calculated as size at maturation divided by age at maturation) and added some brief explanations of how they relate to our choice of traits.

- Figure 4. Although I like the presentation of the results in this way, I would provide some more info in the figure legend, eg. add that is shows the development of the period from 2004 to 2017, based on areas or all stations/trawls? and is it PC1?

Our response: We have added this information to the figure caption.

Appendix C

Dear Editor,

Thank you very much for the opportunity to submit a revised version of our manuscript. We were pleased to see that the reviewers were satisfied by our previous revision. The specific comments provided by reviewer #1 were helpful to further clarify our message, and we provide a point-by-point response below.

Sincerely,
André

--

Referee: 1

Comments to the Author(s).

I think the manuscript has improved, especially regarding the novelty of the study/methods. I understand better the advantage of using the “functional variance” metric, and this elevates the originality of the manuscript. The authors responded well to most of the comments and included some new analyses in supplementary material providing more explanations of the changes occurring in the boreal and arctic regions of the Barents Sea. I am recommending major revision of the manuscript because I still have some misunderstandings about some parts, and some points the authors should work on to finalize their study (developed below). I want to point out that the main part of the manuscript focusing on the CWM/CWV is very clear for me and my remarks focus on the smaller analysis on the functional dispersion calculation and modeling. Authors can find minor comments at the end.

Sampling effort, biodiversity indices & linear model – I am not sure I understand how the taxonomic and functional diversity indices have been calculated. The authors mention the indices “rely on station-level estimates across all years” in their response, and “in each sampling location” L94 in the manuscript. Is each diversity index calculated at the haul level, for each year? Or are hauls somehow aggregated to some station/location/site level? If calculated at the haul level, there is indeed no sampling bias (at least not from the number of hauls) to take into account.

*Our response: Our stations are indeed individual hauls, as described in the methods: trawls towed at 3 kn for 15 minutes.

If calculated at the haul level, I think the subsequent model should be ran using all hauls for all years? I am not sure this was done by the authors, because Figure 2 shows aggregated averages. The authors mention they use averages from sub-regions avoiding a spatial autocorrelation issue (response to reviewer #2). Please be clear on what was done and how: (i) if the model uses aggregates of samples, indices should be corrected for sampling bias using rarefaction curves (ii) if at the haul level, the model has to include spatio-temporal autocorrelation, as mentioned by reviewer #2 (iii) if an analysis is done at a specific scale, it has to be clearly specified in the methods.

If metrics are averaged at the region level before running the linear model, this a bit strange from a statistical point of view. I would have at least preferred something similar to the following study (though diversity metrics not corrected for sampling bias):

-Greenstreet, S. P. R., Fraser, H. M., Rogers, S. I., Trenkel, V. M., Simpson, S. D., and Pinnegar, J. K. 2012. Redundancy in metrics describing the composition, structure, and functioning of the North Sea demersal fish community. – ICES Journal of Marine Science, 69: 8–22.

*Our response: In our analyses we do not aggregate hauls, but take the mean values from each sub-region. Nonetheless, the reviewer is correct that for statistical inference it is better to run the model with all haul values from each sub-region included, but requires to adjust for spatial auto-correlation.

To address the reviewers comment we now provide two complementary analyses: one is based on mean values taken at the sub-region level (already included in the original version of the main manuscript), and another based on all individual hauls in each polygon (added to the SM), with spatial auto-correlation accounted for as in Jørgensen et al. 2019 (MEPS), a paper based on the same sampling design. Despite the larger replication level when using all hauls in the analyses, the two analyses yield the similar results and we have opted for maintaining the analysis and illustration showing only the mean values within sub-region in the main paper.

Linear model – In response to the authors' comments, I don't think that this linear model is proving any causation (do they ever?). I went through the *Laliberte & Legendre 2010* publication: although I do understand how species richness, evenness and functional richness may be linked to functional dispersion, I don't think the 2010 cited paper is particularly inviting to create a model where indices are by construction correlated (since that is what is shown in their simulation study). If authors see a clear benefit to model functional dispersion by other functional diversity indices, they should make sure to justify their choice further. Additionally, if the point is to disentangle the strongest link with richness/evenness/functional richness and functional dispersion (as explained in the authors' response), it needs to be shown in the manuscript. So far, only the coefficient of determination is mentioned, and I could not find any estimate of the explanatory variable effects. Again, these explanatory variables are correlated, and I think that this needs to be evaluated before running the model (variance inflation factor, as mentioned in the previous review).

*Our response: We were interested in disentangling the different contributions to functional dispersion and have update the text in the Results to provide information about each of the three indices included in the model. In the Material and Methods we now provide the motivation for these analyses (L106-109).

L145-147: "In the Arctic, functional richness, species richness and evenness increased faster than in the boreal region but their rate of increase does not explain the higher rate of change in functional dispersion observed in the Arctic (Fig. 2a-d)."

1. Yes, it seems so, and it's a very interesting pattern – but it is not based on any statistical comparison, and could at least include some confidence intervals on the graphs?
2. The 'effects' of evenness/richness/functional richness hypothesized by the authors on the dispersion are not evident from the graphs and could be backed up with the model results?

*Our response: To address comment 1, we now analyse the contribution of the three indices to the trend in functional dispersion by including region-specific linear models. The model results help to address comment 2.

Comparing functional diversity indices – I think it is clear from the analyses conducted that the most insightful one is the CWM-CWV analyses, which also represent the core of the paper. This could be more emphasized in the discussion, giving a perspective on which functional diversity metrics are relevant for studying composition change in communities.

*Our response: Thank you for this suggestion. We have added two sentences to the discussion (L175-179) that highlight the relevance of CWV for community analyses.

Small comments

L39: typo, should be "evenness"

*Our response: Done. Thank you for spotting it.

L40-43: something is wrong with that sentence: "our novel assessment of functional variance indicates that this trend might be short-lived"?

*Our response: We modified the sentence to: “However, the increasing trend observed here may be transitory as the traits involved threaten Arctic species via predation and competition. If the pressure from global warming continues to rise, the ensuing loss of Arctic species will result in a reduction in functional diversity..”

L89-92: “The chosen traits provide information on species characteristics that are involved in assembly processes (habitat affinity and feeding ecology), and ecosystem functions (body size, feeding ecology, and foodweb characteristics) [14].”

Here, it should not be mentioned as ecosystem functions – body size is not a function

*Our response: We indicated examples of effect traits that determine ecosystem function, such as feeding ecology. We have now removed body size because of its indirect relation to ecosystem function.

L103-106:” To disentangle the relative contributions of functional richness, species evenness, and species richness on functional dispersion, we used a general linear model using data from the entire Barents Sea across all years.”

Here again, it’s not really clear what was done, I think more information needs to be detailed on how the data were treated. Is it a model per haul?

*Our response: Please, see response above.

L143-144: “Together, these variables accounted for 44% of the spatio-temporal variation in functional dispersion across the Barents Sea (Fig. S3)”

All explanatory variables need to be mentioned here again for clarity (or in the previous sentence).

*Our response: We added “species richness, species evenness, and functional richness’ to the sentence.

L144 “spatio-temporal variation”: if the model is based on the 2 regions, it’s not really spatio-temporal variation? Reading response to reviewer #2, it’s based on the aggregated regions?

*Our response: The model we refer to here uses data from the individual hauls across the entire Barents Sea, it is not based on aggregated regions (please, see our response above).

L154: remove one “and”

*Our response: We added a “to” that was missing in the sentence: “...links to fish prey, and to bird and mammal predators...”

L196: typo

*Our response: The line number was located in the last paragraph of the previous version of our submission. Unfortunately we could not find which typo this referred to.

Figure S3: please correct typos (“eveness”, “dispesion”) – why is functional richness also called dissimilarity? Dissimilarity measures are usually not richness of a community.

*Our response: Thank you so much for spotting these typos in figure S3. We have now corrected them accordingly. The original figure had ‘functional dissimilarity’ written in one of its axis because of an old terminology we were using in the paper. The measure of functional richness described in the methods and used to produce this figure is based on the sum of the dendrogram branch lengths, which may be seen as a measure of the sum of dissimilarities between species in a community. Thus, the two names refer to the same results. We have now changed Figure S3 to reflect the terminology used in the main paper: ‘functional richness’.

Referee: 2

Comments to the Author(s).

In my opinion the Authors have adressed the Reviewers' comments adequately.

*Our response: Thank you!